# A complex genetic architecture in zebrafish relatives *Danio quagga* and *D. kyathit* underlies development of stripes and spots

**Braedan M. McCluskey**[1], **Susumu Uji**[2], **Joseph L. Mancusi**[1], **John H. Postlethwait**[3], **David M. Parichy**[1,4]*

**1** Department of Biology, University of Virginia, Charlottesville, Virginia, United States of America, **2** Japan Fisheries Research and Education Agency, Watarai, Japan, **3** Institute of Neuroscience, University of Oregon, Eugene, Oregon, United States of America, **4** Department of Cell Biology, University of Virginia, Charlottesville, Virginia, United States of America

* dparichy@virginia.edu

## Abstract

Vertebrate pigmentation is a fundamentally important, multifaceted phenotype. Zebrafish, *Danio rerio,* has been a valuable model for understanding genetics and development of pigment pattern formation due to its genetic and experimental tractability, advantages that are shared across several *Danio* species having a striking array of pigment patterns. Here, we use the sister species *D. quagga* and *D. kyathit,* with stripes and spots, respectively, to understand how natural genetic variation impacts phenotypes at cellular and organismal levels. We first show that *D. quagga* and *D. kyathit* phenotypes resemble those of wild-type *D. rerio* and several single locus mutants of *D. rerio,* respectively, in a morphospace defined by pattern variation along dorsoventral and anteroposterior axes. We then identify differences in patterning at the cellular level between *D. quagga* and *D. kyathit* by repeated daily imaging during pattern development and quantitative comparisons of adult phenotypes, revealing that patterns are similar initially but diverge ontogenetically. To assess the genetic architecture of these differences, we employ reduced-representation sequencing of second-generation hybrids. Despite the similarity of *D. quagga* to *D. rerio,* and *D. kyathit* to some *D. rerio* mutants, our analyses reveal a complex genetic basis for differences between *D. quagga* and *D. kyathit,* with several quantitative trait loci contributing to variation in overall pattern and cellular phenotypes, epistatic interactions between loci, and abundant segregating variation within species. Our findings provide a window into the evolutionary genetics of pattern-forming mechanisms in *Danio* and highlight the complexity of differences that can arise even between sister species. Further studies of natural genetic diversity underlying pattern variation in *D. quagga* and *D. kyathit* should provide insights complementary to those from zebrafish mutant phenotypes and more distant species comparisons.

**Data Availability Statement:** RAD-seq data is available through the National Center for Biotechnology Information Short Read Archive (PRJNA691921). Quantitative trait information,

genotypes, and data used to generate plots are available at https://zenodo.org/record/4685118.

**Funding:** Supported by NIH R35 GM122471 (DMP), NIH NRSA F32 GM119202 and NIH T32-GM07413 (BMM), NIH R01 OD011116 (JHP), and the Japan Fisheries Research and Education Agency (www.fra.affrc.go.jp) (SU). The funders had no role in study design, data collection and analysis, decision to publish, or preparation of the manuscript.

**Competing interests:** The authors have declared that no competing interests exist.

Author summary

Pigment patterns of fishes are diverse and function in a wide range of behaviors. Common pattern themes include stripes and spots, exemplified by the closely related minnows *Danio quagga* and *D. kyathit*, respectively. We show that these patterns arise late in development owing to alterations in the development and arrangements of pigment cells. In the closely related model organism zebrafish (*D. rerio*) single genes can switch the pattern from stripes to spots. Yet, we show that pattern differences between *D. quagga* and *D. kyathit* have a more complex genetic basis, depending on multiple genes and interactions between these genes. Our findings illustrate the importance of characterizing naturally occurring genetic variants, in addition to laboratory induced mutations, for a more complete understanding of pigment pattern development and evolution.

## Introduction

How diverse adult forms arise across species is a classic question in evolutionary biology. Modern evolutionary and population genetic methods have provided insights into the allelic underpinnings of differences between populations and closely related species, yet a fuller understanding of trait evolution requires knowledge of the cellular and developmental mechanisms that translate gene activities into particular morphological outcomes. Achieving such an integrative perspective demands a system accessible to genetic analysis, but also developmental observation and experimental manipulation.

The zebrafish, *Danio rerio* [1], is a major model organism for biomedical research, genetics, and development [2]. With transparent embryos, hundreds of mutants and transgenic lines, and tractability for live imaging, this species has arguably the most accessible development of any vertebrate. Many features are observable even during post-embryonic stages when some new traits arise and some early larval features are remodeled to generate the adult phenotype [3]. One of the most prominent of these traits is the pigment pattern, consisting of horizontal dark stripes of black melanophores and bluish iridophores that alternate with light "interstripes" of yellow-orange xanthophores and gold iridophores [4–7] (**Fig 1**, ***top***). Functional significance of the pigment pattern in nature is not well understood, though both wild and domesticated *D. rerio* attend to pattern variation in choosing shoalmates in the laboratory, and some spotted mutants can be preferred to the wild type [8–10]. Patterns of other teleosts function in mate choice, aggressive displays, avoidance of predation, and other behaviors [11–15]. Other species of *Danio* have stripes, spots, vertical bars and other patterns, and are similar to zebrafish in their amenability to genetic and developmental analyses [10,16–19].

Development of the *D. rerio* adult pattern depends on melanophores and iridophores that arise from post-embryonic neural crest derived progenitor cells within the peripheral nervous system [20–22]. During the larva-to-adult transition, some of these cells migrate to the skin where they differentiate. Some adult xanthophores arise from these post-embryonic progenitors, whereas others develop directly from embryonic / early larval xanthophores that lose their pigment and redifferentiate during adult pattern development [21,23]. Initial patterning of adult stripes depends on cues in the tissue environment that promote the differentiation and localization of iridophores in a primary interstripe in the middle of the flank [3,15,24,25]. Interactions within and between different classes of pigment cells are then required to organize the first two primary melanophore stripes, as well as reiterative secondary and tertiary interstripes and stripes that form as the fish grows [4,6,18,25–33] (**Fig 1**, ***top***).

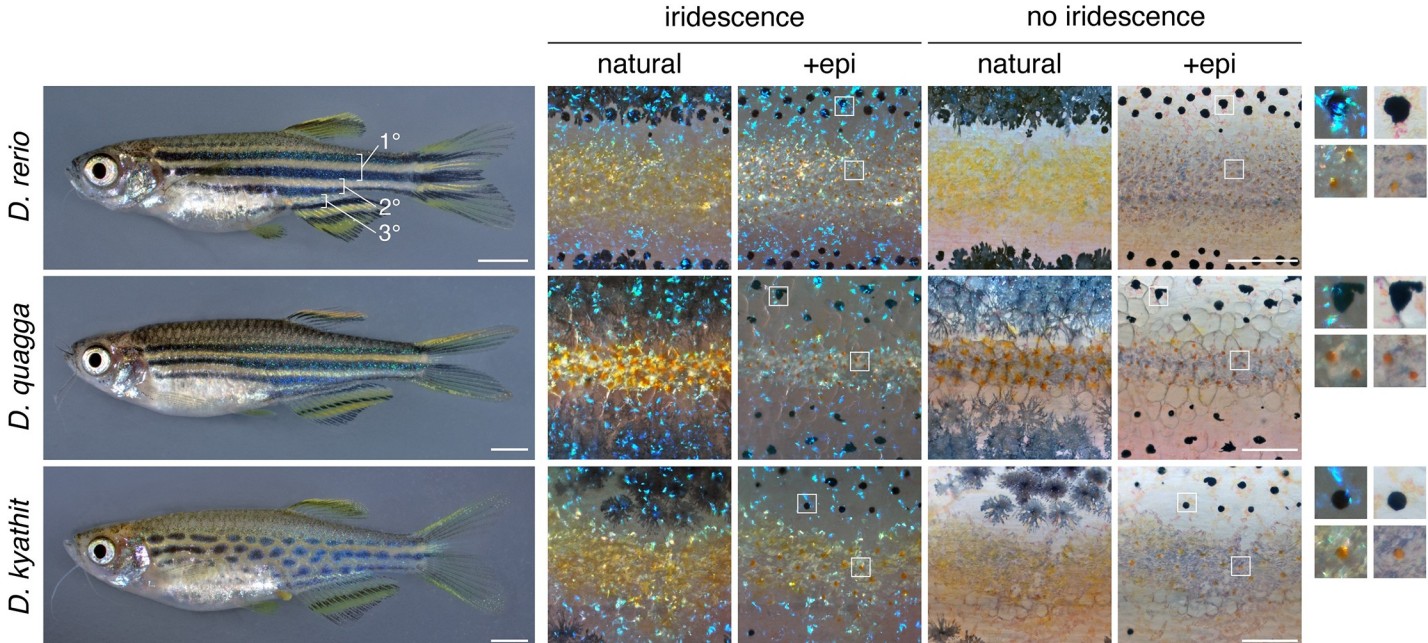

**Fig 1. Pigment patterns and cell types.** *Top*, Striped pattern of *D. rerio* showing primary (1°) light interstripe with primary dark stripes dorsally and ventrally. Secondary and tertiary interstripes and stripes are indicated only ventrally. *Right*, Cells that comprise dark and light pattern elements. Lighting has been adjusted to highlight or eliminate the iridescence of iridophores, and pigment cells are shown both in their natural state and after treatment with epinephrine (epi), which contracts pigment granules of melanophores and xanthophores towards cell centers. Insets at far right show higher magnification views of boxed regions. Dark stripes comprise melanophores with black melanin granules as well as more superficial iridophores that have a bluish iridescence owing to precisely oriented stacks of guanine-containing reflecting platelets (upper insets). Interstripes and interspot regions contain densely packed iridophores with a yellowish hue owing to disordered stacks of reflecting platelets, as well as more superficially located xanthophores, marked by lipid droplets containing yellow–orange carotenoids [6,54]. *Middle* and *Bottom*, Striped and spotted patterns of *D. quagga* and *D. kyathit*. Cell types present in each species were indistinguishable from those of *D. rerio*. In older *D. kyathit* and *D. quagga*, red pigment cells—erythrophores—were present and especially prominent in males, whereas *D. quagga* developed fissures and other disruptions in their stripes, especially in females (S1 Fig). Neither of these features occurs in *D. rerio* and we do not consider them further in this study. Scale bars, 2 mm (left) and 200 µm (right).

The current understanding of adult pigment pattern development in *D. rerio* has been informed by dozens of mutants [4,5,17,23,32,34–43]. Yet it remains unclear if mutants isolated in the laboratory are representative of natural variation within or between species. Most laboratory mutant phenotypes result from loss-of-function mutations in single genes induced by chemical mutagenesis or deliberate targeting of coding sequences. By contrast, standing variation can reflect loss-of-function or gain-of-function alleles, is often polygenic, and results from mutational processes that affect both coding and non-coding sequences.

One powerful approach for discovering naturally occurring genetic variants is meiotic mapping, particularly when combined with second generation sequencing. This strategy has not been feasible so far across *Danio* species, owing to the sterility or gametic aneuploidy of hybrids with zebrafish [44–46]. Nevertheless, crosses between more closely related *Danio* species or divergent populations might yield fertile hybrids, allowing genetic mapping approaches for understanding trait variation in the group, while simultaneously making use of tools and knowledge from the zebrafish system.

Suitable taxa for such an approach are suggested by a genome-scale analysis of *Danio* phylogeny that identified a *D. rerio* species group comprising several closely related taxa having considerable variation in pigment pattern (**Fig 2**) [16]. Among these, *D. quagga* has stripes superficially similar to those of zebrafish ("quagga" refers to the extinct zebra, *Equus quagga*) whereas *D. kyathit* has a pattern of spots ("kyathit" is Burmese for leopard) [47,48] (**Fig 1**).

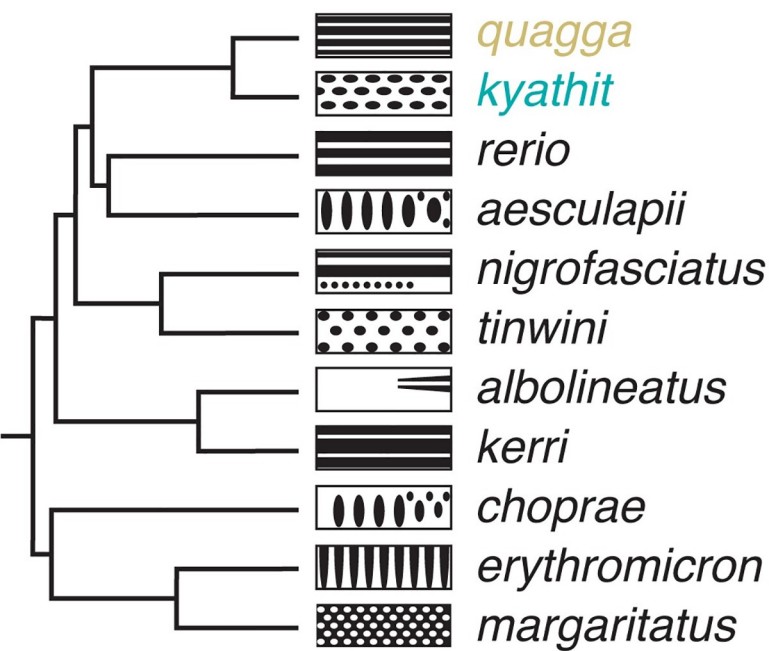

**Fig 2. *Danio* relationships and patterns.** Phylogeny of *Danio* species, recovered in [16], with schematics illustrating adult melanophore pigment pattern variation (left is anterior). *Danio quagga* has also been dubbed *D*. aff. *kyathit* [16,19,47,49] and individuals likely representing the same species were misidentified originally as *D. rerio* [97]. Full color images of live fish can be found in [19] (open access); additional images can be found in [4,16].

Phylogenetically, *D. quagga* and *D. kyathit* are more similar to one another than any other species pair in the genus, but also more different from one another than individuals of any other single species [16]. Geographically, *D. quagga* and *D. kyathit* occur within the Irrawaddy River drainage of Myanmar and are separated from *D. rerio* to the west by the Arakan Mountains. Hybrids between these species and *D. rerio* resemble *D. rerio*, but are infertile [4,49].

In this study we examine the developmental and genetic bases for pigment pattern differences between *D. quagga* and *D. kyathit*. We show that pigment cell complements and phenotypes of *D. quagga* resemble those of wild-type *D. rerio* whereas those of *D. kyathit* resemble some mutant *D. rerio*. We then document differences in pigment cell development leading to striped or spotted phenotypes and show that crosses of *D. quagga* and *D. kyathit* result in fertile hybrids that can be used for genetic mapping. Finally, we utilize quantitative trait locus mapping to test, and reject, the hypothesis that a single major effect locus determines whether stripes or spots form, finding instead that multiple genomic regions contribute to pattern variation segregating in these laboratory crosses. Our analyses reveal the feasibility of quantitative genetic analyses of pigmentation in *Danio* and suggest that exploration of naturally occurring variants should provide insights into the genetics of pigment pattern variation and evolution distinct from those obtainable through laboratory-induced mutant phenotypes of zebrafish.

## Results

### *Danio quagga* and *D. kyathit* pigment pattern phenotypes overlap those of wild-type and mutant *D. rerio*

To better understand the cellular bases of pigment pattern differences between *D. quagga* and *D. kyathit* we first examined complements of pigment cells in young adult fish. We found that

*Danio kyathit* and *D. quagga* had the same classes of pigment cells as wild-type *D. rerio* and these cell types were arranged similarly relative to one another in all three species: dark pattern elements contained melanophores and bluish iridophores and light pattern elements contained xanthophores and gold iridophores (Figs 1 and S1).

Because a proper study of pattern evolution requires a system that can address quantitative differences, we sought metrics that would describe patterns in *Danio* and be robust across species, stages, and lighting conditions. We focused on melanophores because each species has dark pattern elements that include these cells, and because melanophores are more easily documented than xanthophores or iridophores. Moreover, melanophore locations are correlated, or anti-correlated, with the distributions of other pigment cells owing to interactions among pigment cell classes essential for pattern formation [6,7,18,24–26,31,50], suggesting that documentation of just melanophore distributions could define a morphospace useful for describing pattern variation across species in which these interactions and relative cellular arrangements might be conserved. We therefore binarized patterns (melanophore element or not) and defined metrics to represent pattern phenotypes. These included estimates of pattern variation along dorsoventral (DV) and anteroposterior (AP) body axes, and the ratio between these metrics, most easily represented on a log scale, $\log_2$(DV:AP variation). To estimate variation along the DV body axis, we calculated the average grey value for each row (of individually black or white) pixels, representing a single anteroposterior transect across the body. From these average grey values across many rows (transects) in each individual, we then calculated standard deviations, and used these as our estimate of DV variation. We performed the reciprocal calculations to obtain for each individual an estimate of AP variation (i.e., standard deviation of averaged pixel values across DV column transects). Patterns or pattern elements that are nearly uniform along an axis will approach a value of 0, whereas increasing heterogeneity along an axis will yield increasingly large estimates of variation (**Figs 3A** and S2).

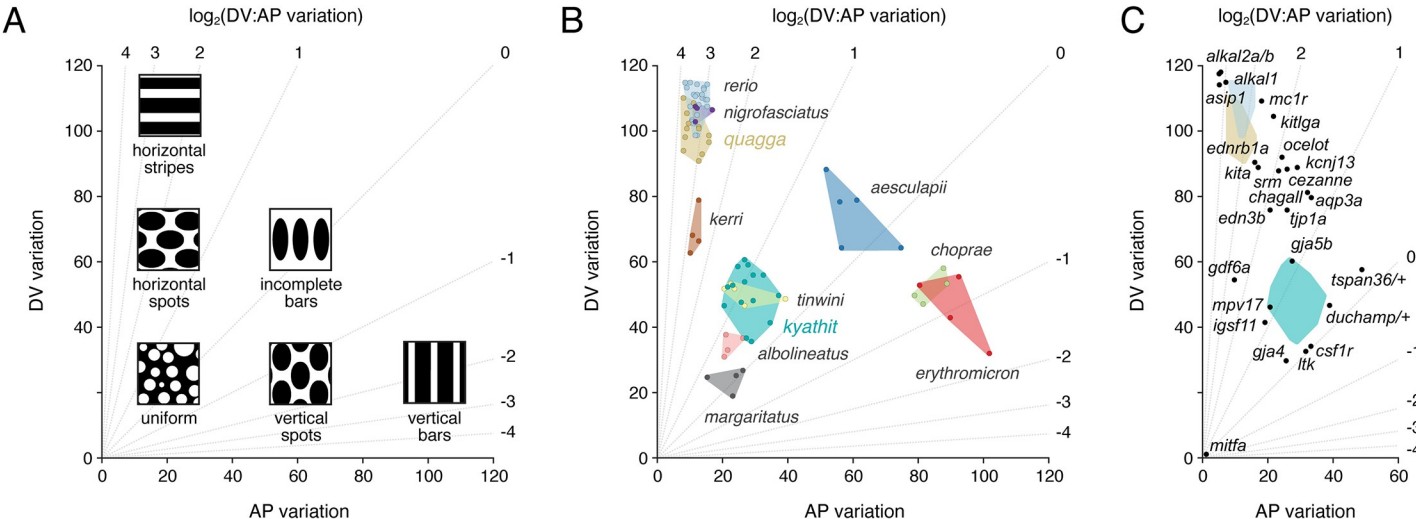

**Fig 3. A morphospace describing pattern variation.** (A) Schematic illustrating where idealized patterns fall in a morphospace defined by DV (dorsoventral) and AP (anteroposterior) pattern variation of melanized elements. Diagonals indicate different $\log_2$ values for the ratio of DV variation: AP variation. (B) Mapping of pattern phenotypes in this morphospace revealed overlap of striped *D. rerio*, *D. quagga*, and *D. nigrofasciatus*, and their difference in morphospace from spotted *D. kyathit* or *D. tinwini*. Additional species occupied distinct regions of morphospace as well. Images of other *Danio* species have been published previously [4,16,19]. Points denote individuals and shapes denote observed phenotypic range. (C) Several single locus mutants of *D. rerio* lie between *D. quagga* and *D. kyathit*, or in the vicinity of *D. kyathit*. These phenotypes arise from null or hypomorphic alleles [22,27, 37,38,42,51,62–64,76,85,86,98–101], though *aqp3a* results from an activating mutation [53], and some mutants have yet to be characterized molecularly (*cezanne, chagall, duchamp, ocelot*) [8,84]. Points for each mutant denote individuals, or averages when images of multiple individuals were available.

We found overlap in pattern morphospace between three striped species, *D. rerio*, *D. quagga*, and *D. nigrofasciatus*, and two spotted species, *D. kyathit* and *D. tinwini* (**Fig 3B**). A species with three relatively broad and diffuse stripes, *D. kerri*, fell in between, whereas a species with particularly small and symmetrical light spots on a dark background, *D. margaritatus*, was separated from the other spotted danios. Species with vertical bars—*D. aesculapii*, *D. choprae* and *D. erythromicron*—occupied a distinct region of morphospace with higher AP variation. Finally, a species with a mostly diffuse, nearly uniform pattern, *D. albolineatus*, mapped close to *D. kyathit*. This latter grouping, in which two visually disparate patterns nevertheless occupied similar positions in morphospace, suggests that additional metrics ultimately will be needed to describe the broader range of phenotypes across the genus.

This pattern morphospace also allowed us to consider single locus pigment pattern mutants of *D. rerio*, representing a range of patterns, some of which have spots rather than stripes. Many of these mutants were situated either between *D. rerio* or *D. quagga* and *D. kyathit*, or, in the case of some spotted mutants, in the vicinity of *D. kyathit* (**Fig 3C**). These findings suggest the hypothesis that patterns of *D. quagga* or *D. kyathit* are separated by just a single mutational step, and identify *a priori* candidate genes that might contribute to the difference between these species.

## Differentiation timing and morphogenesis differ during ontogeny of striped *D. quagga* and spotted *D. kyathit*

To understand the ontogeny of pattern differences between *D. quagga* and *D. kyathit*, we followed individual cell behaviors across daily image series in individual fish from larval through late juvenile stages. In comparison to *D. quagga*, melanophore numbers in *D. kyathit* increased more rapidly, and these cells reached higher total numbers (Fig 4A and 4B). This situation implies that spots in *D. kyathit* are not simply a consequence of fish having insufficient numbers of melanophores to fill a striped pattern; this was surprising in comparison with spotted mutants of *D. rerio*, in which melanophores are often fewer (e.g., *ltk*, *gja5b*, *igsf11*) [24,39,51] or of similar abundance (*tjp1a*, *aqp3a*) [52,53] to that of wild-type. In contrast to *D. quagga*, melanophores of *D. kyathit* were also more densely packed, achieving lower nearest-neighbor distances and higher densities within melanized pattern elements overall (Figs 4C and S3A). As pattern formation proceeded, new melanophores appeared as lightly melanized cells unpaired with other cells (Fig 4A), consistent with *de novo* differentiation from unmelanized precursors as occurs in *D. rerio*, rather than division of existing melanophores [23,54]. Melanin contents of melanophores, as inferred from two-dimensional areas of contracted melanin granules [54], did not differ between species during pattern ontogeny but were ultimately greater in adult *D. kyathit* than *D. quagga* (S3B Fig).

Melanophore patterns emerged gradually. In both *D. quagga* and *D. kyathit*, a primary interstripe of iridophores formed near the horizontal myoseptum, followed by dorsal and ventral primary melanophore stripes (**Fig 4A**). In *D. quagga*, secondary melanophore stripes then appeared, but in *D. kyathit*, primary stripes subdivided into spots and additional spots emerged dorsally and ventrally, generating a much larger number of melanized pattern elements overall (**Fig 4D** and **S1 Movie**). Ontogenetic changes in pattern were also evident in DV and AP variation metrics (S3C **and** S3D **Fig**) and in the ratio of these metrics (**Fig 4E**). Although total areas of the flank covered by melanized pattern elements did not differ during larval stages of adult pattern formation, *D. kyathit* adults had less total area covered by melanized pattern elements than *D. quagga*, despite the former having more melanophores (**S3E Fig**). The higher density of melanophores achieved by *D. kyathit* was manifested at the level of individual cell behaviors, as development of spots in this species was accompanied by more

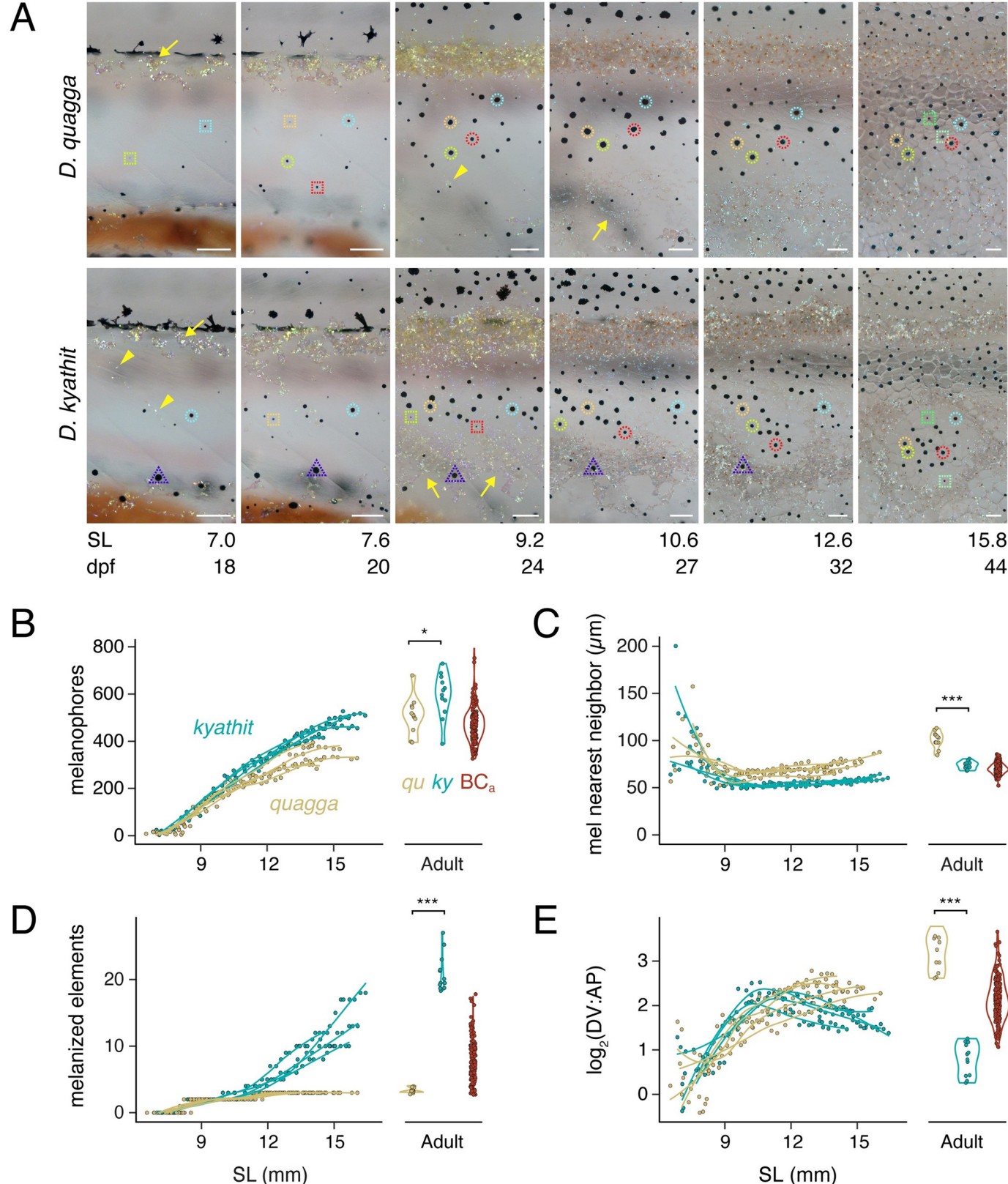

**Fig 4. Pigment pattern development in *D. quagga* and *D. kyathit*.** (A) Representative individuals imaged repeatedly during adult pattern formation. Images are aligned to show corresponding regions and rescaled to control for overall growth (frames and regions are selected from S1 Movie). Standard length (SL in mm)

serves as a proxy for developmental stage as the relationship between development rate and days post fertilization (dpf) depends on rearing conditions [3]. By 7.0 mm SL, adult melanophores had started to develop in both species. Newly differentiating melanophores are marked by dashed squares, and the same cells are marked by dashed circles of the same color in subsequent images. Adult melanophores ultimately coalesced into dark pattern elements with movements by individual cells and rearrangements among cells particularly apparent in *D. kyathit*. As in *D. rerio*, melanophores were occasionally lost in both species (e.g., dashed triangle in *D. kyathit*) [25,99]. Densely arranged iridophores of the primary interstripe were evident by 7.0 mm SL in both species. Subsequently developing iridophores of stripes or spots (yellow arrowheads) appeared earlier in *D. kyathit* (7.0 mm SL) than *D. quagga* (9.2 mm SL), as did iridophores of secondary interstripes or "interspots" further ventrally (yellow arrowheads; 10.6 and 9.2 mm SL, respectively). To facilitate cell counting, fish were treated prior to imaging with epinephrine to contract melanin granules towards cell centers. Images shown are representative of 4 individuals of each species imaged throughout the stages of early adult pigment pattern development. (B–E) Pattern metrics during development of fish imaged repeatedly (splines over points), and in young adults and backcross progeny (most 22–25 mm SL, 4–5 months post-fertilization; see main text and S3 Fig). Individual points for melanophore nearest neighbor distances (C) represent median values calculated for all melanophores examined within each individual fish at time points with ten or more melanophores. *, $P = 0.029$ in B ($F_{1,21} = 5.52$); ***, $P < 0.0002$ in C ($F_{1,21} = 19.83$), $P < 0.0001$ in D ($F_{1,21} = 421.04$), $P < 0.0001$ in E ($F_{1,21} = 217.11$). Scale bars, 100 μm.

pronounced movements of melanophores than were observed in *D. quagga* (S4 Fig; S1 Movie).

Locations of melanophores in *D. kyathit* and *D. quagga* presumably depend on interactions with iridophores and xanthophores as in *D. rerio*: interstripe iridophores promote the localization of melanophores in stripes and stimulate the development of interstripe xanthophores, which themselves help to organize melanophores; additional iridophores differentiate within stripes but are not known to be essential for stripe formation [6,24–26]. In both *D. quagga* and *D. kyathit*, onset of adult pattern development was marked by differentiation of iridophores in a nascent primary interstripe near the horizontal myoseptum (Fig 4A), as in *D. rerio*. Subsequent iridophore development differed between species, however, as new iridophores outside of the primary interstripe—contributing to dark pattern elements (stripes or spots) and additional light pattern elements (interstripes or "interspots")—appeared earlier in *D. kyathit* than *D. quagga* (S6 Fig). Densely arranged interspot iridophores of *D. kyathit* developed over a broader area of the flank than interstripe iridophores of *D. quagga*, and as these cells differentiated, light interspot elements appeared in regions initially harboring only melanophores (S1 Movie). The inability to reliably distinguish individual iridophores in brightfield images precluded quantifying their numbers.

Adult xanthophores first became apparent at similar stages in the two species [S6A Fig; standard length (SL) of first appearance, $F_{1,6} = 2.36$, $P = 0.1754$]. During ontogeny, interstripe elements of *D. quagga* and interspot elements of *D. kyathit* became populated with xanthophores, though these cells were somewhat fewer and less densely arranged in *D. quagga*. By adult stages, however, xanthophores in *D. quagga* were somewhat more numerous than in *D. kyathit* (S3F Fig). Visible xanthophore pigment was similar between species during larval through juvenile stages but slightly greater in *D. kyathit* adults than *D. quagga* adults (S3G Fig).

Finally, because dynamics of body growth relative to pigment cell development could affect pattern [55,56], we examined two measures of body dimensions: standard length (SL) and height of the body at the anterior margin of the anal fin (HAA) [3]. During pattern ontogeny, *D. kyathit* were ~6% larger in both dimensions than *D. quagga* but adult sizes were not significantly different from one another (S3H and S3I Fig). Though faster early growth could, in principle, cause a pattern to be broken into smaller elements (e.g., stripes becoming spots), the ~20% greater complement of melanophores in *D. kyathit* compared to *D. quagga*, and differences in melanophore morphogenesis as well as iridophore and xanthophore development, make it unlikely that differences in body size or growth trajectory play major roles in determining pattern differences between these species.

Together, these findings reveal several distinct cellular phenotypes associated with spotted and striped patterns (Table 1) consistent with species differences depending on variation at

**Table 1. Qualitative summary of phenotypes.** Interpretations derived from numerical and statistical analyses in Figs 4, S3, S4, S6 and S1 Movie.

| | *D. quagga* | *D. kyathit* |
|---|---|---|
| melanophore number | ++ | +++ |
| melanophore "packing" (density, inverse of NN distances) | + | ++ |
| melanized elements | + | +++ |
| $\log_2$(DV:AP) variation | +++ | + |
| melanized element coverage of flank | +++ | ++ |
| melanophore movement | + | ++ |
| xanthophore number | +++ | ++ |
| early iridophore appearance outside of primary interstripe | – | + |

one or a few highly pleiotropic loci, or perhaps a larger number of loci each having more specific effects.

## Genetics of pigment pattern variation in *D. quagga* and *D. kyathit*

Morphometric analyses showed that *D. quagga* and *D. kyathit* have phenotypes similar in morphospace to wild-type zebrafish and some single locus mutants of zebrafish, respectively (Fig 3C), suggesting the hypothesis of a simple genetic basis for the pattern difference between species. The phylogenetic proximity of *D. quagga* and *D. kyathit* [16] raised the possibility that interspecific mapping crosses could be used to test this hypothesis. We therefore generated F1 hybrids, finding them to be robust and fertile, and to have pattern phenotypes more similar to *D. quagga* than *D. kyathit* (Fig 5A). We then backcrossed F1 hybrids to *D. kyathit* to assess genetic correlations among phenotypes and to map genomic regions associated with pattern variation for establishing a lower bound on the number of contributing loci.

Using a family of backcross progeny (BC$_a$, $n$ = 158; **Fig 5A**), we examined correlations among metrics described above and additional metrics, which revealed four principal components (PCs) that explained ~73% of total phenotypic variance (**Fig 5B**). PC1 was most closely associated with overall pattern, as individuals having high scores in PC1 resembled *D. quagga* in overt phenotype and higher $\log_2$(DV:AP variation) and DV variation, whereas individuals with low scores in PC1 resembled *D. kyathit* with higher AP pattern variation and melanized pattern element count. PC2 was associated with melanophore and xanthophore spacing and overall density, PC3 with melanophore element coverage and inferred melanin content, and PC4 with a deeper body (ratio of HAA to SL) in females and a narrower body in males.

To learn whether one or several loci contribute to pattern variation in this cross, we used quantitative trait locus (QTL) mapping. Parental fish were not inbred, so this approach will identify chromosomal regions associated with variation between *D. quagga* and *D. kyathit*, and also variation segregating within the *D. kyathit* stock to which F1 hybrids were backcrossed (**Fig 6A**). We genotyped siblings by RAD-tag sequencing and isolated 2,458 phased, polymorphic markers flanking SbfI restriction sites across the genome [57,58]. We then mapped these sites to the *D. rerio* genome using parameters to account for sequence divergence between closely related species [59].

These analyses revealed several genomic regions that contributed to phenotypic variation, allowing us to reject the hypothesis that a single major effect locus is responsible for the species-level transition between stripes and spots. Three QTL were linked to variants present in the hybrid parent, representing alleles that differ between the *D. quagga* and *D. kyathit* backgrounds: melanophore nearest-neighbor distance mapped to a broad region of Chr15 (peak LOD = 6.8; $q$<0.05; **Fig 6B** and **6E**); $\log_2$(DV:AP variation) mapped to a broad region of

A

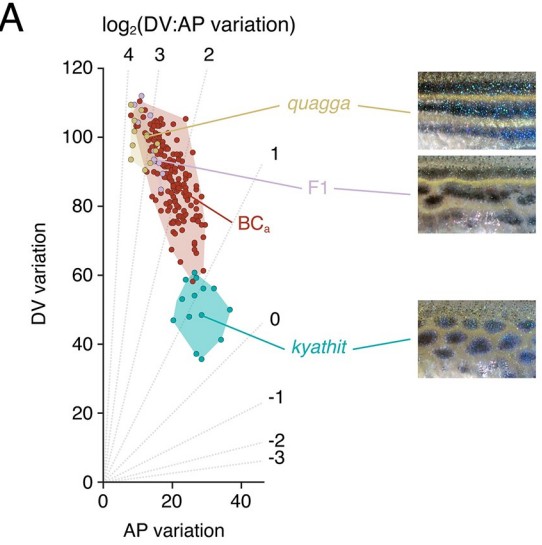

B

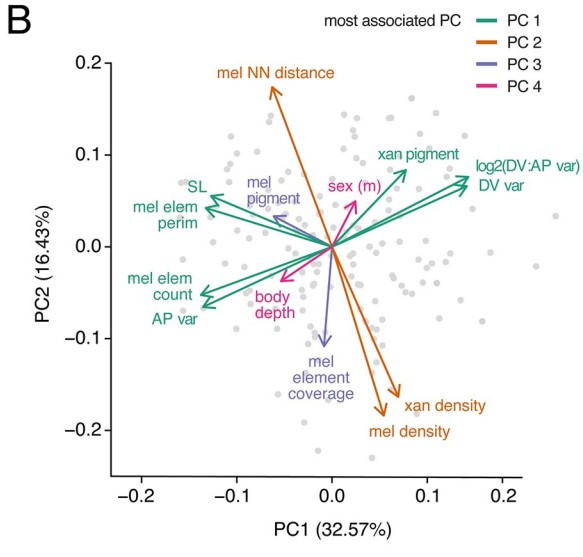

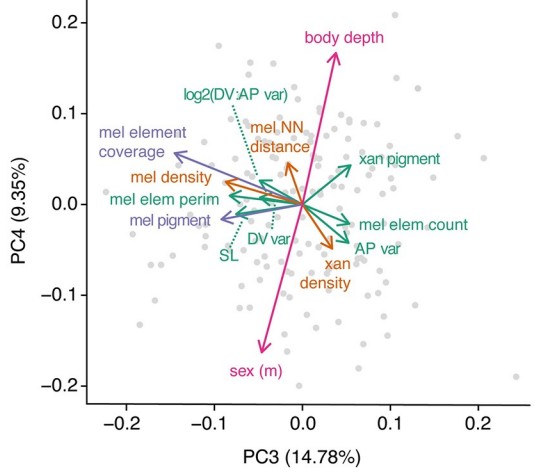

**Fig 5. Hybrid and backcross progeny phenotypes of *D. quagga* and *D. kyathit*.** (A) Pattern variation morphospace showing locations of parental *D. quagga* and *D. kyathit*, F1 hybrids, and backcross progeny (BC$_a$). Images of *D. quagga* and *D. kyathit* patterns are details of the same individuals shown in Fig 2 and the F1 hybrid shows a typical pattern in this cross. (B) Principal components (PC) of phenotypic variation in BC$_a$ progeny with percent variance explained noted in parentheses. Melanized element count and perimeter refer to entire spots or stripes. Melanized element coverage refers to total area covered by melanized elements relative to total imaged flank area. Likewise, melanophore and xanthophore densities were calculated relative to total area. Melanophore and xanthophore pigment refer to areas of contracted pigment granules, rather than entire cell sizes. For sex, arrow points towards values typical of males. NN, nearest neighbor.

Chr18 (peak LOD = 4.7 with marginal significance at $q<0.05$; **Fig 6C** and **6E**); and PC3—associated with melanized pattern element coverage and melanin content—mapped to Chr23 (peak LOD = 5.0; $q<0.05$; **Fig 6D** and **6E**). Four additional QTL were linked to variants segregating within the *D. kyathit* background (**Figs 6E** and **S6**). These were associated with variation in melanophore density (Chr8), melanized area per cell (Chr8), and xanthophore density (Ch11). An additional QTL (Chr20) was associated with log$_2$(DV:AP variation), DV variation, number and perimeter of melanized pattern elements, and PC1 (**Figs 6C** and **6E** and S6).

Several of these chromosomal regions harbor known pigment pattern genes. For example, the very broad region associated with log$_2$(DV:AP variation) between *D. quagga* and *D. kyathit* on Chr18 includes *slc24a5* and *mc1r*, though functions for these genes in melanophore differentiation are not obviously relevant to pattern element shape [60,61]. A region associated with several metrics on Chr20 includes *kita*, required for melanophore development, whereas the region associated with PC3 on Chr23 includes *edn3b* and *alkal2a*, required for iridophore development and thus indirectly for melanophore pattern [17,20,22,39,62–64]. Spontaneous alleles of *kita* and *edn3b* or their homologues have been found affecting pigmentation in natural or domesticated populations of other species [65–70], though these analyses tend to argue against a major effect role for *kita* between *D. kyathit* and *D. quagga*, as variation identified at this locus segregated within the *D. kyathit* background rather than between *D. kyathit* and *D. quagga*.

Among phenotypes with significant peaks, log$_2$(DV:AP variation) was associated with the genotype inherited from the hybrid sire on Chr18 and the *D. kyathit* dam on Chr20. To determine if these regions represented variants with additive effects or an epistatic interaction, we compared individuals with each of the four possible genotype combinations (**Fig 7A**, *upper plot*). Three genotype combinations had phenotypes midway between the two parental species whereas one combination was associated with striped patterns more similar to *D. quagga*, indicating a strong epistatic interaction between these QTL. This epistasis was manifested in morphospace as a shift upward toward the region occupied by *D. quagga* and other striped species (**Fig 7B**). Epistasis between Chr18 and Chr20 extended to other phenotypes as well. For example, after accounting for the minimum three stripes, individuals with both stripe-associated alleles averaged about half as many additional melanized pattern elements as siblings with other genotypes (~2.8 vs. ~5.5, respectively; **Fig 7A**, *lower plot*).

Finally, to further assess roles for inter-species QTL on Chr15 and Chr18, and to potentially identify other QTL, we generated independent F1 hybrids and a second backcross family (BC$_b$) to *D. kyathit* (**Fig 6A**). To account for any parent-of-origin effects, we used a hybrid female in BC$_b$ as opposed to the hybrid male used in BC$_a$. We reasoned that major-effect loci should be identifiable by genotyping backcross progeny that most resembled the parental phenotypes. Extreme sampling approaches can use fewer individuals while offering similar power as larger scale experiments that incorporate many uninformative intermediate phenotypes [71]. We therefore generated 457 full siblings and selected individuals representing opposite ends of the pattern spectrum ($n$ = 18 striped and $n$ = 18 spotted, 3.5% at each extreme)

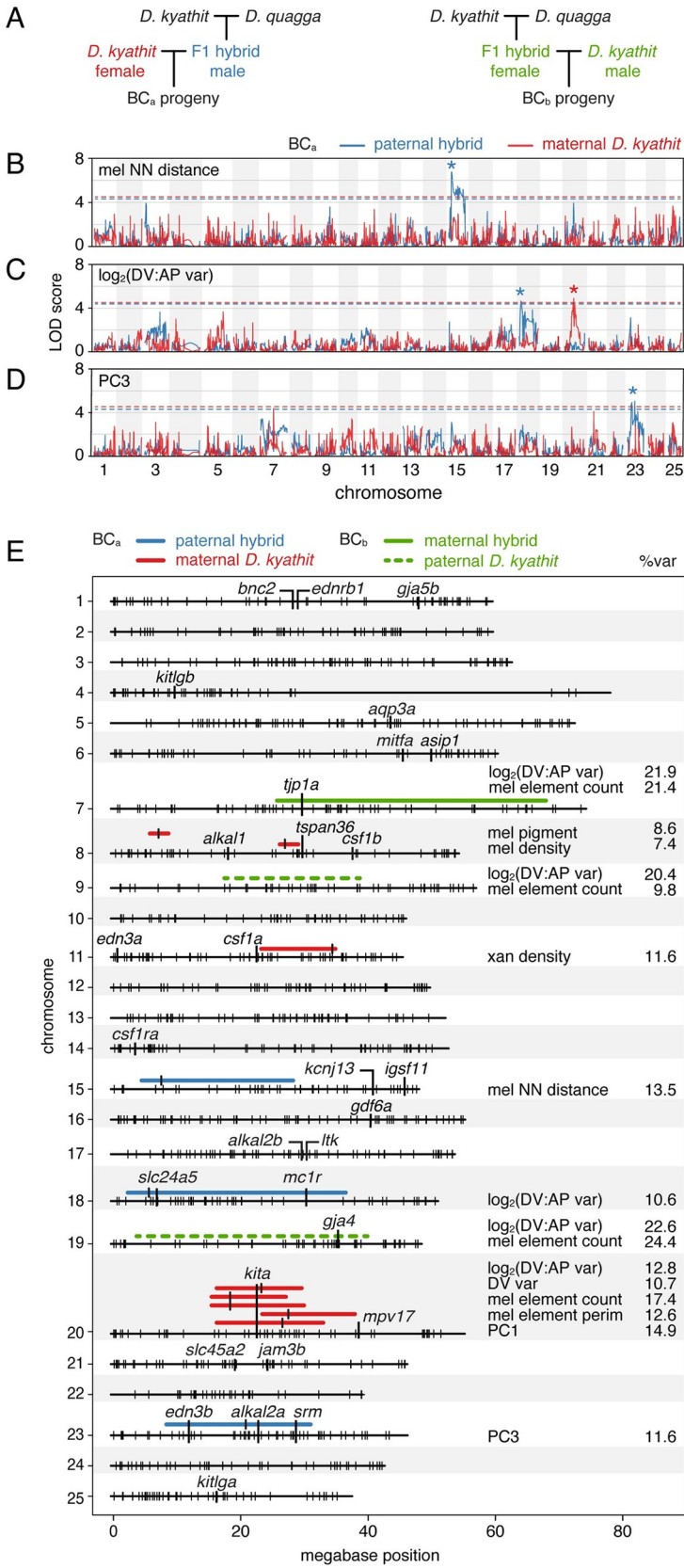

**Fig 6. Chromosomal regions associated with pattern and size metrics in backcross individuals.** (A) Crossing schemes for mapping variation in two independent families, BC$_a$ and BC$_b$ (see main text). (B–C) Log odds scores of example QTL (BC$_a$) for melanophore spacing (nearest neighbor distance, log$_2$(DV:AP variation), and number of melanophore elements across 25 chromosomes. Variation segregating between species in blue and within *D. kyathit* in red. Dashed lines indicate $q = 0.05$ false discovery thresholds and asterisks indicate peak LOD scores. (E) Summary of regions associated with pattern variation and size in the same family referenced in B–C (BC$_a$), and defined by $F_{ST}$ values for phenotypically extreme individuals from a second family (BC$_b$; see main text). Regions highlighted have peaks ($q<0.05$, LOD$\geq$4) marked by vertical bars and widths defined by positions 1.5 LOD lower than peaks. Percents of phenotypic variance explained by QTL (%var) are indicated at right. Small, vertical hash marks on chromosomes denote BC$_a$ maternal *D. kyathit* marker loci and positions of several genes having roles in pigment pattern formation in *D. rerio* are indicated as well.

allowing calculation of fixation indices, $F_{ST}$, across the genome to assess relative allele frequencies between and within phenotypic classes (**Fig 7C and 7D**).

Remarkably, analyses of BC$_b$ individuals failed to recover any of the genomic regions identified in the first cross and instead revealed broad segments of elevated $F_{ST}$ on Chr7, Chr9, and Chr19 (**Figs 6E** and S6). Inspection of haplotypes revealed that alleles driving the extreme $F_{ST}$ values were transmitted both by the hybrid dam (Chr7) and the *D. kyathit* sire (Chr9 and Chr19), indicating that both interspecific and intraspecific variants affected pattern in this second backcross, as in BC$_a$. QTL-linked variants on each of these three chromosomes affected both log$_2$(DV:AP variation) and melanophore element count (**Figs 7C** and **7D** and S8). All

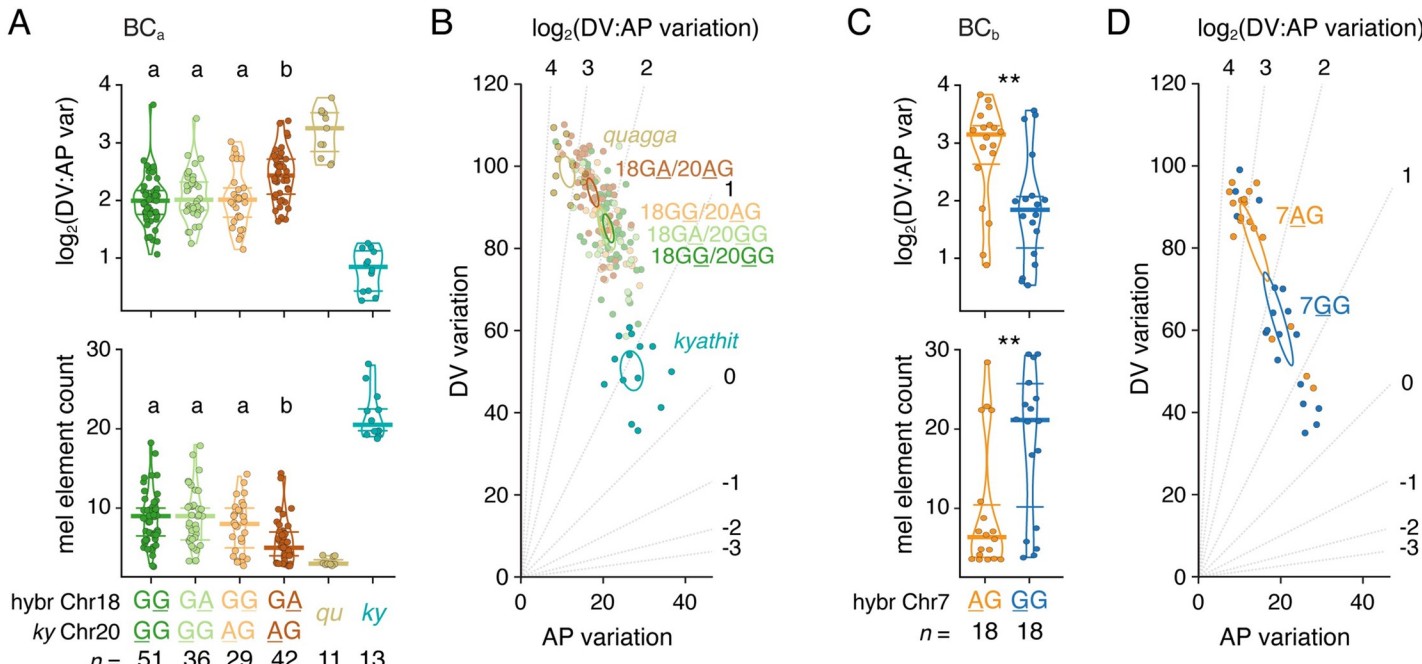

**Fig 7. Epistatic relationships between QTL influence pattern phenotype.** (A) Genotypes at Chr18:5246981 and Chr20:23025453 linked to variants affecting log$_2$(DV:AP variation) and melanophore element count in BC$_a$ progeny. Genotypes are listed as maternally-inherited *D. kyathit* allele followed by paternally inherited allele with phenotype-associated alleles underlined. Bars indicate medians ± interquartile range. Overall analyses of variance for allelic combinations in BC$_a$ progeny: *upper*, $F_{3,154} = 7.26$, $P<0.0001$; *lower*, $F_{3,154} = 10.07$, $P<0.0001$. Means not significantly different from one another ($P>0.05$) in *post hoc* Tukey-Kramer comparisons are indicated by shared letters above data points. (B) Three combinations of Chr18 and Chr20 alleles overlapped in morphospace. When striped alleles on Chr18 and Chr20 were together, a phenotype significantly more similar to that of *D. quagga* developed. Ellipses indicate 95% confidence intervals of means. (C) Genotype at Chr7:27412951 impacts log$_2$(DV:AP variation) and melanophore element count in BC$_b$ progeny selected for striped and spotted phenotypes. Analyses of variance: *left*, $F_{1,34} = 9.30$, $P = 0.0044$; *lower*, $F_{1,34} = 8.91$, $P = 0.0052$. (D) Morphospace position was impacted by variants linked to maternally inherited *D. kyathit* alleles at Chr7:27412951. Most siblings with the A allele had patterns similar to *D. quagga* whereas progeny with the G allele were more similar to *D. kyathit*. (compare with representative parental species values in A). Statistical analyses in A and C used DV:AP variation and *ln*(melanophore elements), which stabilized among-group differences in residual variance evident in original values.

three stripe-associated alleles co-occurred in 8 of 18 striped individuals, whereas three spot-associated QTL co-occurred in 10 of 16 spotted individuals suggesting that our extreme phenotype sampling approach selected for combinations of multiple variants impacting pattern.

Though genes previously associated with pigment pattern were not found within the region of high $F_{ST}$ on Chr9, near the peak $F_{ST}$ values on Chr7 and Chr19 were located *tight junction protein 1a* (*tjp1a*) and *gap junction protein alpha 4* (*gja4*), respectively. The products of these genes interact physically and *D. rerio* mutants of each have spotted phenotypes [52,72] (**Fig 3C**), making them viable candidates for contributing to the naturally occurring and complex genetics of pattern variation across *D. quagga* and *D. kyathit*.

## Discussion

We identified developmental and cellular features of pigment pattern formation in striped *D. quagga* and spotted *D. kyathit* and demonstrated that differences between these species have a complex genetic architecture. During early adult pattern development, both species formed a rudimentary striped pattern, similar to *D. rerio*. Subsequently, however, melanophores of *D. quagga* moved little, whereas melanophores of *D. kyathit* migrated into spots as iridophores and then xanthophores differentiated around them. Species differences in overall pattern and cellular characteristics that arose during this time were maintained into young adulthood, including shape and arrangement of pattern elements (as captured by metrics of DV and AP pattern variation), and the relative numbers of melanophores and distances between them. Adults also differed in later-arising characteristics of inferred melanin content per cell, total coverage by melanized pattern elements, and numbers of xanthophores. In backcross hybrids, variation in several pattern and cellular attributes assorted independently and mapped to distinct chromosomal regions. Some phenotypes were associated with allelic differences between *D. quagga* and *D. kyathit* (e.g., melanophore nearest neighbor distance on Chr15), others were associated with alleles segregating within *D. kyathit* (melanophore pigment on Chr8), and others had interspecific and intraspecific components at different locations [$\log_2$(DV:AP variation) on Chr18 and Chr20]. A complete understanding of pattern variation in these species will thus need to account for evolutionary alterations at multiple loci, potentially having direct or indirect effects on the differentiation and morphogenesis of multiple cell types.

By studying natural phenotypic variation, our study uncovered complexities of pigment pattern evolution in a way that cannot be achieved solely from laboratory-induced mutations. We found that in *D. rerio*, single loss-of-function mutations were sufficient to move the pattern through a substantial region of a morphospace defined by dorsoventral and anteroposterior pattern variation. Yet, the loss-of-function phenotypes most often isolated in forward or reverse genetic screens are unlikely to represent the full spectrum of naturally-occurring allelic variation influencing pigment cells and the patterns they form. Pattern mutants isolated in the laboratory are typically chosen and recovered because they have marked and consistent pattern defects. Alleles of small effect are more likely to be missed, and these, as well as alleles that are difficult to characterize molecularly (e.g., mutations affecting non-coding sequences) may be less likely to be isolated. By contrast, naturally occurring variation might, or might not, be under selection, and alternative alleles would be expected to have varying degrees of penetrance and expressivity. Moreover, our analysis of the broader morphospace occupied by *Danio* species (**Fig 2**) illustrates that some phenotypes—like vertical bars—appear to be relatively inaccessible in *D. rerio*, even after decades of genetic screening.

In focusing here on naturally occurring variation in *D. quagga* and *D. kyathit*, we were able to find several QTL affecting phenotypes of hybrid, backcross individuals. The QTL we identified are notable for being several in number, in some instances epistatic to one another, and

different between crosses derived from different parents. Some of the associated regions harbored genes known to have roles in pigment pattern formation in *D. rerio* (e.g., *tjp1a*, *gja4*, *kita*) that are plausible candidates for variation in *D. quagga* and *D. kyathit*. Nevertheless, some QTL corresponded to genomic regions not previously associated with pigmentation or pigment pattern in *D. rerio*. That such regions were identifiable, even with mapping crosses having modest numbers of individuals, suggests that further efforts to identify segregating variation within and between species of *Danio*, combined with mapping or association studies at larger scale, could provide mechanistic and evolutionary insights complementary to what can be learned from zebrafish.

Simple genetic bases for some naturally occurring pigmentary differences have been found in other systems [69,73–75]. Yet an important conclusion of our study is that pattern variation in *D. quagga* and *D. kyathit* is complex developmentally and genetically. Like some *D. rerio* mutants, *D. kyathit* initially formed two primary stripes but these broke into spots as development proceeded. This failure of stripe integrity was not the result of a single alteration, for example the absence of a single class of pigment cell, or to pleiotropic changes owing to mutation at a single, major effect locus, as can be the case for mutants of *D. rerio* [20,23–28,43,52,76]. Our findings are, however, concordant with more complex genetic architectures that have been found to underlie naturally occurring pigment pattern variation in some other systems. In beach mice, for instance, three loci of the agouti signaling pathway contribute to a white phenotype [77]. The main variant, which affects ligand binding efficiency, is not fixed across populations with similar phenotypes. In pigeons, coloration has complex epistatic interactions involving three loci [78] and pattern within individual feathers has an additional, independent genetic component [79]. By extension, differences in pigment pattern between other pairs of *Danio* species presumably also have polygenic bases, as has been noted previously [4,17,18,80]. Though some genes likely contributing to such variation have been identified, additional genes undoubtedly await discovery. In *D. quagga* and *D. kyathit*, the recovery of multiple, different loci in independent crosses demonstrates a polygenic and still-evolving genetic basis for the transition between stripes and spots.

A number of developmental genetic mechanisms have been proposed to explain the evolution of pigment patterns within *Danio* and teleosts more generally [15]. For example, species-differences in body size at the onset of pigment pattern formation, and differences in dorso-ventral growth, are believed to influence adult patterns of several species within *Danio* and the closely related genus *Devario* [55]. We found that *D. quagga* and *D. kyathit* initiated adult pattern development at a similar body size and had similar growth trajectories, arguing against causal roles for heterochronic shifts or allometric alterations at the whole-organism level. Nevertheless, our observations are consistent with finer-scale heterochronic changes contributing to pattern differences: *D. kyathit* first developed iridophores outside of the primary interstripe at an earlier stage than *D. quagga*, and a more exuberant expansion of iridophore-containing "interspot" elements was associated with the formation of melanophore spots rather than stripes. Given the interactions among pigment cell classes required for pattern formation (revealed by studies of *D. rerio* [4]), changes in the time or place at which any one class appears could have cascading effects. For example, earlier and broader development of xanthophores is associated with a nearly uniform distribution of melanophores in *D. albolineatus*, a phenotype that can be recapitulated transgenically in *D. rerio* [18]. Here, differences in iridophore development might drive whether stripes of spots form, much as changes in iridophore development lead to spots in *tjp1a* mutant *D. rerio* [52] and a reduced number of stripes in *D. nigrofasciatus* [17]. Alternatively, changes to the network of interactions among pigment cells could themselves contribute, as inferred for some *D. rerio* mutants [28,29], and as simulated in models of pattern formation [35,81,82]. Indeed, the quantitative data we provide should help

to parameterize agent-based models of pigment pattern development. Our integrative approach illustrates the value of analyzing natural variation at both phenotypic and genotypic levels for better understanding pigment pattern evolution and development beyond the laboratory.

## Materials and methods

### Ethics statement

All animal research was conducted according to federal, state and institutional guidelines and in accordance with protocols approved by the Institutional Animal Care and Use Committee of University of Washington (4094–01) and the Animal Care and Use Committee of University of Virginia (4170). Anesthesia and euthanasia used MS-222.

### Fish stocks and rearing conditions

*D. quagga* and *D. kyathit* were obtained from the pet trade and reared subsequently in the same conditions and recirculating water system used to maintain *D. rerio* (~28˚C; 14L:10D). Though DNA sequences of *D. quagga* type specimens have not been described, fish used in this study were morphologically indistinguishable from published photographs of *D. quagga* [48] (**S1A Fig**). Fish were crossed by pair-wise natural matings to produce larvae of each species for analysis, or F1 progeny, which were then backcrossed to other *D. kyathit* to produce crosses for genetic mapping (BC$_a$ and BC$_b$). Adult fish were anesthetized prior to imaging and fin clipping for subsequent DNA extraction. Live *D. rerio* used for imaging (**Fig 1**) or morphospace analysis (**Fig 3C**) were wild-type strain NHGRI-1 [83], *cezanne^utr17e1^* [84], *chagall^vp36rc1^* [84], *edn3b^vp30rc1^* [17], *duchamp^utr19e1^* [49], *igsf11^utr15e1^* [51], *kita^b5^* [62], *leo^t1^* [76] *ltk^j9s1^* [85], *mitfa^w2^* [38], *ocelot^vp37rc1^*, *tspan36^wpr21e1^* [27]. Phenotypes of additional *D. rerio* mutants were analyzed from published images: *alkal2a^ya340^*, *alkal2a^ya342^* [64], *aqp3a^tVE1^* [53], *asip1* [42], *gdf6a^s327^* [37], *gja4* [29], *kitlga^tc244b^* [22], *srm^t26743^* [86], *tjp1a^twl1^* [52].

### Phenotyping of adults and longitudinal imaging during development

Anesthetized fish were imaged using a Zeiss AxioObserver inverted microscope (BC$_a$), a Zeiss AxioZoom stereo zoom microscope, or a Nikon D810 digital single lens reflex camera with 105 mm Nikkor macro lens as appropriate for the size of the fish (BC$_b$). Images used for BC$_a$ and daily image series were recorded after first treating fish with 10 mM epinephrine to contract pigment granules towards cell centers. Morphometrics (standard length, height at pelvic fin, etc.) were measured in ImageJ. Images were then isometrically scaled and aligned in Adobe Photoshop 2018 to ensure that subsequent analyses used the same region across fish and developmental stages. The region of interest was centered slightly anterior to the vent and extended dorsoventrally to encompass the three melanophore stripes in a mature adult zebrafish. Regions of interest were exported as RGB images and melanophore location and size were measured in ImageJ [87] across eight body segments. For repeated imaging of pigment pattern development, the region of interest was truncated anteroposteriorly (relative to adult images) to allow for cell tracking and quantification at earlier developmental stages.

To segment melanophores from RGB images, we identified objects of the appropriate color and size after removing lighting artifacts and background. To reduce background variability from lighting artifacts such as reflection and iridescence from iridophores, we removed bright outliers within a 25-pixel radius. We then isolated the red channel to better distinguish black melanophores from xanthophores and the underlying skin and muscle (which has a light pink color). We then applied automatic local thresholding using the Sauvola method (originally

designed to detect numerous black text characters on a light page) with radius set to 25 and the k parameter set to 0.34. Segmented objects with area between 100 and 1500 pixels were retained as melanophores. Yellow/orange xanthophores with high red values and low blue values were similarly segmented after subtracting the red channel from the blue channel, using the Sauvola method with the same parameters, and selecting objects between 10 and 200 pixels. Improperly segmented cells including uncontracted melanophores were manually corrected in ImageJ and replaced with circles of diameter 15 pixels. Melanophore nearest neighbor distances were calculated using the spatstat package in R [88].

For measurements of melanophore movement, segmented melanophores from eight daily image series ($n$ = 4 *D. quagga*; $n$ = 4 *D. kyathit*) spanning 18 to 45 dpf were used. Melanophores were converted to points of uniform size to allow image registration optimization in ImageJ using the Linear Stack Alignment with SIFT plugin with a rigid transformation, inlier ratio of 0.20, and default parameters for all other values. Aligned melanophores were tracked using the TrackMate v5.2.0 plugin [89] in ImageJ allowing for up to 100 px displacement over 24 hours. Cells tracked across seven or more days with no more than one missing day were included in analyses.

Pigment patterns were quantified using binary images with each pixel segmented according to whether it was part of a melanophore pattern element. Melanophore elements of epinephrine-treated fish were defined as all pixels within 20 pixels of the center of a melanophore identified using the previously described segmentation parameters. This distance approximates the average distance between nearest neighbor melanophores measured previously. To obtain binarized pigment patterns for fish not treated with epinephrine, regions of interest were thresholded manually to contain similar amounts of segmented pixels as epinephrine-treated images. A number of images, mostly drawn from previous publications describing species or mutants, required manual segmentation due to image compression for publication or uneven lighting. Pattern morphospace location was defined by horizontal and vertical variation, which were calculated as the standard deviation of the 8-bit gray value profiles along the respective axis of each binary image. When possible, images of multiple individuals were used to account for phenotypic variability present within species or between individuals mutant for the same locus.

First appearance of adult xanthophores was determined by inspection of color brightfield images of daily image series for the presence of orange carotenoids pigmentation, evident in fish that had been treated with epinephrine. Appearances of iridophores outside of primary stripes were recorded when sparsely arranged iridophores were first evident in brightfield micrographs within the prospective ventral primary stripe and when first evident where near the ventral edge of the flank where prospective ventral secondary interstripe or interspot elements first form.

For display, images were color-balanced and levels adjusted in Adobe Photoshop 2020. In Fig 1, backgrounds of whole fish were partially desaturated for color to better represent the natural appearance of the pigment pattern.

## Genomic DNA extraction and genotyping

DNA was extracted from adult fin clips using a DNeasy 96 Blood & Tissue Kit (Qiagen) or a MagAttract HMW DNA kit (Qiagen). DNA concentrations were measured with a Qubit 2.0 fluorometer and standardized prior to genotyping. Fish for each mapping cross were genotyped using RADseq. The BC$_a$ cross used single digest RADseq with high fidelity SbfI (NEB) as previously utilized with D. kyathit [16]. Libraries were sequenced on two high output lanes of an Illumina NextSeq using paired end 75 bp reads. The BC$_b$ cross employed double digest

RADseq with high fidelity PstI (NEB) and EcoRI (NEB) as described previously [90] and was sequenced on one lane of an Illumina HiSeq with paired end 150 bp reads. Reads were aligned to the zebrafish genome (GRCz10) using the Burrows-Wheeler Aligner and 19-base kmers (bwa mem -k 19). Reads with multiple best mapping locations and reads mapping to annotated repetitive elements were excluded from further analyses. Mapping to the zebrafish genome allowed the use of existing annotations for inferring genes likely to be present within intervals defined by QTL. This approach is reasonable in the present context as the karyotype of 25 pairs of metacentric or submetacentric chromosomes is largely conserved across diploid species within Cyprinidae, which includes genus *Danio*, and comparison of zebrafish to common carp *Cyprinus carpio* revealed largely conserved intrachromosomal structure between even these divergent taxa [91–93]. Nevertheless, testing of individual candidates identified in this manner warrant further validation of candidate gene linkage with QTL as well as finer scale analysis of synteny relationships across these species. Genotypes were called using the ref_map.pl script from the Stacks pipeline with default parameters [94,95]. Genomic regions in the $BC_b$ cross with loci having significantly different genotype frequencies were identified from $F_{ST}$ values calculated by the Stacks pipeline from genotype calls.

### QTL mapping and statistical analyses

For the $BC_a$ cross, principal components of the mappable phenotypes were calculated in R using prcomp() from the 'stats' package. Genotypes of progeny were phased based on parental genotypes using Joinmap (Kyazma). Loci with any missing genotypes were excluded from analyses. QTL mapping was then performed using the 'qtl2' package in R [96]. Phased genotypes were mapped separately based on parent of origin in the $BC_a$ cross. QTL were identified by Haley-Knott regression using the scan1(), scan1perm(), and find_peaks() functions in the 'qtl2' package. QTL windows were defined using a 1.5 LOD drop from the most-associated marker.

Parametric and non-parametric analyses of other quantitative data were performed in JMP 14 (SAS Institute, Cary NC).

### Supporting information

**S1 Fig. Morphometrics and pattern features of older adult *D. quagga* and *D. kyathit*.** (A) Body proportions of striped fish used in this study were more similar to proportions described originally for *D. quagga* than *D. kyathit*, based on the small numbers of preserved specimens on which original species descriptions were based (*n* = 5 and *n* = 6, respectively) [47,48]. Shaded regions in plots indicate approximate proportions as inferred from published reports. Points and bars (median ± interquartile range) represent adult fish (*n* = 14) sampled from stocks used in this study. head, head length; pec, pectoral fin length; preD, pre-dorsal fin length. (B) In adult *D. quagga*, stripes are initially uniform (Fig 2) but develop fissures and reticulations after several months as fish continue to grow. These are particularly evident in deeper bodied females (upper left, ~35 mm SL). In *D. kyathit*, the early adult pattern is maintained during later stages. (C) Details of female *D. quagga* in A, illustrating stripe reticulations (left) and, in the boxed region, red erythrophores (arrow, right). Scale bar, 2 mm in B. (TIF)

**S2 Fig. Image processing and morphospace assessment.** Whole-fish images were measured for size morphometrics, then isometrically scaled based on opercle and caudal peduncle and aligned so the primary interstripe was horizontal. The region of interest (ROI) was isolated and glare of iridophores removed followed by isolation of the red channel alone for visualizing

melanophores in grey scale. Melanophores, defined by spots of contracted melanin pigment, were then segmented and thresholded. Additional cropping and filtering for size maxima and minima were applied for assessing cellular level metrics. The same approach was applied to xanthophores using gray scale images derived from the blue and red channels. For global pattern, melanized regions were expanded and then blurred to fill gaps, allowing for overlap similar to true cell edges (Fig 1 "natural"), and a global intensity threshold applied, defining melanized elements from which pattern metrics could be extracted directly (count, area, perimeter, relative coverage) or after isolating and averaging DV and AP grey value profiles. (TIF)

**S3 Fig. Pigment pattern and size during development and in adults.** Additional quantitative metrics of patterns representing melanophore spacing and melanized pattern element distribution (A–E), xanthophore number and pigment (F,G), and body size (H,I). (A) Melanophore densities were greater in *D. kyathit* than *D. quagga*, though densities overall fell during development as fish grew in size. (B) Melanin content of individual melanophores did not differ during early adult pattern ontogeny but was significantly greater in older adult *D. kyathit* than *D. quagga*, as inferred from two-dimensional area of contracted melanin granules [54]. Values shown are medians for all melanophores quantified within an individual fish. (C) DV pattern variance continued to increase in both species as rudimentary primary stripes become increasingly organized and variation continued to increase in *D. quagga* as secondary stripes were added dorsally and ventrally. In *D. kyathit*, however, melanophores initially in primary stripes, as well as melanophores further dorsally and ventrally, clustered into spots, causing a progressive reduction in DV variation as some dorsoventral transects came to have little coverage by melanized pattern elements. (D) AP variation initially increased in both species, but then fell in *D. quagga* as anteroposteriorly oriented stripes became more orderly. AP variation increased in *D. kyathit* as initially continuous, rudimentary stripes were broken into spots, such that anteroposterior transects crossed spot–interspot boundaries. (E) The percent of the flank covered by melanized pattern elements was initially similar between species, though adults of *D. quagga* ultimately had greater coverage than *D. kyathit*. (F,G) Xanthophores first became visible later in development than melanophores and were ultimately more numerous in *D. quagga* than *D. kyathit*, though having slightly more visible pigment in the latter. (H,I) During development, *D. kyathit* were slightly larger than *D. quagga* (least squares means, SL: 11.8 vs. 11.1 mm, pooled SE = 0.03; HAA: 2.0 vs 1.9 mm, pooled SE = 0.01) after controlling for individual variation. Adult sizes were not significantly different. ***, $P<0.0001$ in A ($F_{1,21} = 74.15$), C ($F_{1,21} = 315.28$), D ($F_{1,21} = 104.00$), E ($F_{1,21} = 89.72$); **, $P = 0.0002$ in B ($F_{1,21} = 19.47$); $P = 0.0038$ in F ($F_{1,21} = 11.66$); *, $P = 0.038$, in G ($F_{1,21} = 4.89$); ns, not significant. (TIF)

**S4 Fig. Melanophore movements during pattern formation were greater in *D. kyathit* than *D. quagga*.** (A) Tracks indicate displacements of melanophores within regions shown in Fig 4A during the course of repeated imaging. (B) Total displacements and median speeds per day of melanophores observed during repeated imaging (pooled data for $n = 4$ larvae of each species). Horizontal bars indicate medians and interquartile ranges. As time points were isometrically scaled to allow for alignment and fish growth, all measurements are in scaled pixels. Analyses of variance for species differences, after controlling for nested (random) effects of individuals within species: displacement, $F_{1,6} = 15.11$, $P = 0.0074$; median speed, $F_{1,6} = 7.15$, $P = 0.0366$. Original values were ln-transformed and square root-transformed, respectively, to control for heteroscedasticity of residuals. (TIF)

**S5 Fig. Iridophore and xanthophore development.** (A) After iridophores of the primary interstripe (e.g., arrow) developed, xanthophores began to differentiate nearby (insets, 7.7 mm SL), similar to *D. rerio*. Later, iridophores that will contribute to melanized pattern elements began to differentiate in *D. kyathit* (inset and arrowheads, 8.1 mm SL) and became more numerous thereafter, but iridophores had not yet appeared at these stages in *D. quagga*. (B) Sizes at which iridophores of stripes or spots or secondary, ventral interstripes or interspots were first evident in *D. quagga* ($n = 7$) and *D. kyathit* ($n = 6$) imaged daily through these stages of pigment pattern formation. Bars show medians with quartiles and observations from the same individuals are connected by lines. Species difference in multivariate analysis of variance, $F_{1,11} = 22.64$, $P = 0.0007$. Scale bars, 100 μm.
(TIF)

**S6 Fig. Genetic mapping of adult trait variation in *D. quagga* and *D. kyathit*.** Chromosomal regions associated with example pattern and size metrics, sex, and principal components (PC) 1–4 in BC$_a$ progeny. Plots shown in Fig 6B–6D are shown here as well for ease of comparison. Asterisks indicate QTL exceeding 5% false discovery thresholds (dashed lines).
(TIF)

**S7 Fig. Chromosomal regions contributing to pattern variation across *D. quagga* and *D. kyathit* revealed by genome-wide $F_{ST}$ scan.** (A) Brackets indicate chromosomal regions with several loci having $F_{ST}$ values exceeding an adjusted $P$-value of 0.01 (dashed line) and highlighted in Fig 6E. Isolated markers exceeding the threshold of statistical significance (e.g., Chr5) likely represent false-positives, because these analyses used full siblings and pattern-associated variants are physically linked to other variants; given the high marker densities achieved in these analyses, singleton peaks are more likely to reflect errors in mapping or geno-typing. (B) Striped and spotted siblings used for $F_{ST}$ scan.
(TIF)

**S8 Fig. Effects of Chr9 and Chr19 QTL in BC$_b$ progeny.** Panels show effects of alleles linked to variants at Chr9:36449721 (A,B) and Chr19:26824025 (C,D). Analyses of variance, in A: *upper*, $F_{1,34} = 6.44$, $P = 0.0159$; *lower*, $F_{1,34} = 2.95$, $P = 0.0951$; in C, *upper*, $F_{1,34} = 7.57$, $P = 0.0094$; *lower*, $F_{1,34} = 8.47$, $P = 0.0063$. Other annotations and analysis as in Fig 7.
(TIF)

**S1 Movie. Dynamics of pattern formation in *D. quagga* and *D. kyathit*.** The movie shows representative individuals of *D. quagga* and *D. kyathit* imaged over 30 d of adult pattern formation (selected regions and days of these same individuals are shown in Fig 4A. Images are rescaled to control for overall growth. In *D. kyathit*, note especially the incursion of dense "interspot" iridophores into regions initially containing melanophores, and subsequent movement of melanophores into spots.
(AVI)

## Acknowledgments

Thanks to Emily Bain and Larissa Patterson for assistance with imaging, and Amber Schwindling and Marianne Cole for help with fish crosses and fish rearing.

## Author Contributions

**Conceptualization:** Braedan M. McCluskey, Susumu Uji, John H. Postlethwait, David M. Parichy.

**Data curation:** Braedan M. McCluskey, David M. Parichy.

**Formal analysis:** Braedan M. McCluskey, Susumu Uji, Joseph L. Mancusi, David M. Parichy.

**Funding acquisition:** Braedan M. McCluskey, Susumu Uji, John H. Postlethwait, David M. Parichy.

**Investigation:** Braedan M. McCluskey, Susumu Uji, Joseph L. Mancusi, David M. Parichy.

**Methodology:** Braedan M. McCluskey.

**Project administration:** David M. Parichy.

**Resources:** David M. Parichy.

**Software:** Braedan M. McCluskey.

**Supervision:** David M. Parichy.

**Validation:** David M. Parichy.

**Visualization:** Braedan M. McCluskey, David M. Parichy.

**Writing – original draft:** Braedan M. McCluskey, David M. Parichy.

**Writing – review & editing:** Braedan M. McCluskey, Susumu Uji, John H. Postlethwait, David M. Parichy.

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
