## [Decision Letter · Decision Letter 0]

19 Mar 2021

Dear Dave,

Thank you very much for submitting your Research Article entitled 'A complex genetic architecture in zebrafish relatives Danio quagga and D. kyathit underlies development of stripes and spots' to PLOS Genetics.

The manuscript was fully evaluated at the editorial level and by independent peer reviewers. The reviewers appreciated the attention to an important topic but identified some minor changes to improve further the clarity of your manuscript, that we ask you address in a revised manuscript.

We therefore ask you to modify the manuscript according to the review recommendations. Your revisions should address the specific points made by each reviewer.

[LINK]

Yours sincerely,

Robert N. Kelsh

Guest Editor

PLOS Genetics

Gregory Barsh

Editor-in-Chief

PLOS Genetics

This is a detailed, thorough and well-illustrated study, examining the genetic basis for naturally-evolved pigment patetrn differences in an experiemtnally accesible model system. Parichy and colleagues take advantage of phylogenetic proximity of two Danio sp. with distinct pigment patterns, and which form fertile hybrids, to begin to dissect the basis for natural pigment pattern evolution, in the context of the experimental studies in zebrafish which examine single locus mutations affecting coding regions and which generate large effects.

They begin by developing a quantitative measure of Danio pigment pattern, enabling ready comparison of spotted v striped patterns. Whilst in many ways this simply confirms the obvious visual differences seen in the 3 species, it has the advantage of giving an objective measure of the pattern differences, albeit restricted to melanophores, for both the DV and AP axes, and so comes into its own when examining rerio mutants and quagga x kyathit hybrids. The description in the Results of why they do this (p.4 Results second paragraph) is clear, but the description would benefit from slight expansion to explain the metrics used to define melanophore pattern; I am struggling to pin down what exactly the DV:AP ratio is measuring. The S3 Fig is helpful for explaining how images are processed, but doesn’t quite explain this final step.

Subsequently they perform a characteristically thorough and highly quantitative assessment of pigment pattern formation in both species. This section is very well documented, but somewhat overwhelming as presented. A summary table clarifying the observed similarities and distinctions in the evolution of pattern in the two species, with direct comparison to WT (and mutant?) zebrafish, would be helpful in highlighting clearly to the reader which features are candidates for being relevant to the switch between spots and stripes.

F1 hybrids between the 2 species were generated and patterns quantitated for comparison with parents, before backcrossing to kyathit for genetic characterisation by QTL mapping. Two independent backcrosses identify broad candidate regions on multiple chromosomes associated with a set of pigment pattern differences between the two species, although none were shared between the two backcrosses. None of these are directly characterised, but several are associated with known rerio pigmentation genes; however, the linked domains are very extensive, so it remains unclear if these known loci are contributing.

The authors main conclusion is that natural pigment pattern evolution in these two species appears to result from a complex genetic mechanism, involving multiple loci, similar to observations in other species.

Minor point:

p.11 ‘This situation contrasts with spotted mutants of D.

rerio, in which melanophores are often fewer (e.g., ltk, gja5b, igsf11) [23, 38, 50] or of similar

abundance (tjp1a, aqp3a) [51, 52] to that of wild-type.’ For this comparison will be useful, it seems that comparison of quagga and hyathit melanophore density to WT zebrafish is required.

Reviewer's Responses to Questions

**Comments to the Authors:**

Reviewer #1: Nicely conducted, and clearly reported, study. I have no significant concerns, as the data collection, and statistical analysis, have been carried out to a high standard. The general conclusion - that natural variation has a different, and more complex genetic basis than discovered in laboratory mutants - is not surprising to me, but is still not generally appreciated. This paper lays out a quantitative methodology that should be broadly applicable to other studies/systems.

Just a few minor typos remaining, for example:

Fig 6. Chromosomal regions associated with pattern and size metrics in F2 backross individuals. (backcross)

Reviewer #2: In their manuscript "A complex genetic architecture in zebrafish relatives Danio quagga and D. kyathit underlies development of stripes and spots", McCluskey and co-authors provide an extremely thorough analysis of the genetic, cellular and developmental underpinnings interspecies variation in pigmentation pattern in Danio species.

The authors first document the pigmentation phenotypes of the three Danio species, Danio rerio, Danio quagga (both striped) and Danio kyathit (spotted) at a cellular level. Next, they quantified dorso-ventral and anterior-posterior variation of melanic patterns across the Danio species complex and in mutant lines of the zebrafish, Danio rerio. This allowed the authors to compare the pattern variation by presenting the data in a 2-dimensional morphospace. The rest of the analyses focuses on the two sister species Danio quagga (striped) and Danio kyathit (spotted). A detailed analysis of pigment pattern development in the two species shows how the two phenotypes diverge during development. An analysis of the genetic basis using quantitative trait loci mapping provides insights into the (very polygenic) genetic basis of the phenotypic variation in pigmentation patterns between these two species.

I very much liked this manuscript for three reasons:

i. The study investigates a phenotype (pigmentation patterns) from three completely different angles. It investigates the phenotypes from an evolutionary viewpoint by comparing pattern across species, provides a very detailed analysis of the developmental differences between two sister species that differ in their pigmentation and lastly and most importantly gives insights into the genetic basis underlying this variation. It therefore provides a uniquely complete and integrative analysis.

ii. The analyses conducted in this study have a very high quality and are very elaborate.

iii. I could imagine that it might be initially disappointing to realize that the identification of target genes (or even mutations) turned out to be so difficult because of the complex genetic basis. Nevertheless, I found the take-home message a very important one: That the genetic basis of natural variation is often much more complex, even if phenocopies could be generated by a single mutation.

I would like to congratulate the authors to their work. I have only a few, mostly minor suggestions.

Additional comments:

1) I initially thought that the F2 have been generated through inbreeding of F1 individuals. I think it would be helpful to indicate earlier that those are backcrosses. Also, a crossing scheme could be helpful, especially to more easily follow the results in Fig. 6D that are very complex. Also, the method/concept of QTL mapping could be maybe already introduced earlier (in the introduction; for example, in the paragraph “One powerful approach …” or in the last one).

2) I enjoyed reading the introductory paragraph of the species. One information that would be nice to have there too, is the divergence time between Danio quagga and Danio kyathit and both to Danio rerio, also to know what “closely related” means in this fish family.

3) Although the schematics are cute and helpful, I would consider adding actual photographs to Figure 2 on top. Also, seeing photographs of some of the Danio rerio mutants that are close to the D. kyathit morphospace would be a helpful visualization of one of the key messages (one could get there with a single mutation, too). Please ignore the request in case it is difficult because of copyright issues.

4) While I found Figure 5 to be really interesting to understand the correlation between the cellular characteristics and pigmentation quantifications. This is now only a suggestion, but I kept wondering where the parental species and F1s would be in this PCA. If measurements for these are available a helpful visualization would be to perform a PCA with the parental species only and then project the F1s and BC individuals on the original PCA.

5) It would be great if the authors could expand a bit more on the method part of the QTL (how was the linkage map generated, how were markers filtered). Also, is there any evidence for larger rearrangements between these two species and the Danio rerio genome that could have affected the analysis (i.e., where linkage map and marker position on the D. rerio genome compared)?

6) In case genomic sequences of Danio quagga and Danio kyathit are available: Are there any coding differences between the target genes within the target intervals in Fig. 6 (tjpa, gja4, slc45a, kitla, …). I assume the responsible mutations are rather non-coding, but as comparing the coding sequences might be relatively straight-forward it might be worth checking.

Reviewer #3: McCluskey and colleagues have submitted a manuscript describing an approach to quantify morphometric features for stripe/spot patterns in closely related danio fish species. They then used this quantification scheme to map QTL's of two closely related species of danio, one with stripes the other with spots that when intercrossed produced fertile offspring. They were able to determine regions on several chromosomes that had significant lod scores resulting in their being able to reject the single gene hypothesis for pattern differences. Further selective crosses identified other loci with variation both between species and within each species.

Honestly the paper is well written, clear, the data of high quality. I have no significant critiques for improving the paper.

minor:

-fig 2 is not referenced in the text.

-I do wonder if the significant regions could have been narrowed rapidly and more finely by doing low coverage whole genome sequencing on pools of classified offspring looking for regions with loss of hetrozygosity. Not required for the paper, just curious.

Reviewer #4: In this manuscript, McCluskey et al. make use of two closely-related Danio species with divergent pigment patterns to begin explore the genetics underlying these differences. This approach relies upon the ability to generate fertile hybrids of the two species, D. quagga and D kyathit, and a morphometric approach to quantify various elements of the adult pattern for the purpose of identifying QTL’s. The study represents an alternative to the standard genetics of the Danio rerio model, by interrogating the nature of phenotypic differences in natural populations as opposed to those generated via laboratory-induced mutations.

The data in the paper are complex but presented with great clarity and rigor. Although the basic result that the stripe vs. spot difference between the two species is polygenic, and that while several genomic regions of interest are identified there are no “smoking guns” might be seen as slightly disappointing, in fact it validates the approach. This result is of great importance as biologists increasingly try to expand beyond just the traditional model systems.

My only suggestions:

Fig 5A presents shows a typical example of the F1 hybrid pattern, but I’d also be curious to see examples of the extremes in the backcross generation used for the Fst scan (perhaps added to S8 Fig). Also, are the individual male and female hybrids used for the backcrosses represented here? It would be interesting to know where each lies in the morphospace.

If I understand correctly, the differences observed between the results of the two backcrosses (Figs 6 and 7) are likely due to interspecies variation as opposed to the sex of the parent-of-origin per se having some functional role? Perhaps that could just be clarified with a sentence somewhere.

Small points:

p. 5 fourth line from the bottom: word “adult” is repeated

p. 6 last line before “Genetics of pigment pattern variation” subheading: “each having more specific effects”

S1 Fig legend third line from bottom: presumably should be “~3.5 cm SL”, not 3.5 mm

Reviewer #5: The manuscript by McCluskey et al. describes their approach to identify genetic bases determining ‘stripes’ or ‘spots’ adult pigment pattern in Danio species. Their study is very intriguing and challenging: they used two close relatives of Danio species which can produce fertile F1 progeny, thus enabling them to use F2 for mapping natural genetic variants between the two species, responsible for the different pigment patterns of ‘stripes’ or ‘spots’. Unfortunately, their genetic analyses haven’t reached a clear conclusion: instead of finding a simple main effect locus involved, they revealed a complex genetic basis with several loci contributing to pattern variation. In overall, the results are convincing and the strategies described are valuable for future research by others.

Comments:

Their genetic analyses pointed out some loci, where the candidate genes known to be involved in pigment pattern formation are located. Are any of these responsible for the formation of stripe or spot?

They don’t mention about the difference across species in sequence there, nor in their expression (special, temporal or level). Are there any nucleotide alterations? If any, are they all non-coding?

Their imaging analyses for chromatophore behavior upon pigment pattern formation in the two species revealed different timing of melanophore differentiation is key to generate distinct pattern, ‘stripes or spots’, suggesting that heterochrony should explain for this interspecies difference. Are the above loci related to the heterochrony?

‘Morphospace’ part is hard to understand, especially Fig 5B. Where is Fig 5C?

In Fig. 7, is A consistent with B in genotypes? Is the maternal allele always indicated first (left)? Why are the genotypes different between 7A and 7B?

SL appears prior to spelling-out (page 6, 2nd and 3rd paragraph).

Duplicate ‘adult’ (page 5, last paragraph).

**Have all data underlying the figures and results presented in the manuscript been provided?**

Reviewer #1: Yes

Reviewer #2: **No: **No, but the authors have declared that data will be available upon publication through public repositories.

Reviewer #3: Yes

Reviewer #4: Yes

Reviewer #5: Yes

PLOS authors have the option to publish the peer review history of their article (what does this mean?). If published, this will include your full peer review and any attached files.

Reviewer #1: No

Reviewer #2: **Yes: **Claudius Kratochwil

Reviewer #3: **Yes: **Shawn Burgess

Reviewer #4: No

Reviewer #5: No

---

## [Editor Report · Decision Letter 1]

8 Apr 2021

Dear Dr Parichy,

We are pleased to inform you that your manuscript entitled "A complex genetic architecture in zebrafish relatives Danio quagga and D. kyathit underlies development of stripes and spots" has been editorially accepted for publication in PLOS Genetics. Congratulations!

Yours sincerely,

Robert N. Kelsh

Guest Editor

PLOS Genetics

Gregory Barsh

Editor-in-Chief

PLOS Genetics

Comments from the reviewers (if applicable):

Dear Dave

Thanks for addressing the comments in detail, and of rsubmitting to PloS Genetics.

Best

Robert

**Data Deposition**

http://datadryad.org/submit?journalID=pgenetics&manu=PGENETICS-D-21-00047R1

**Press Queries**

---

## [Editor Report · Acceptance letter]

20 Apr 2021

PGENETICS-D-21-00047R1 

A complex genetic architecture in zebrafish relatives Danio quagga and D. kyathit underlies development of stripes and spots 

Dear Dr Parichy, 

We are pleased to inform you that your manuscript entitled "A complex genetic architecture in zebrafish relatives Danio quagga and D. kyathit underlies development of stripes and spots" has been formally accepted for publication in PLOS Genetics! Your manuscript is now with our production department and you will be notified of the publication date in due course.

With kind regards,

Katalin Szabo

PLOS Genetics

On behalf of:
